# Defect-driven selective metal oxidation at atomic scale

Qi Zhu [1,6], Zhiliang Pan[2,6], Zhiyu Zhao[1,6], Guang Cao[1], Langli Luo [3], Chaolun Ni[1], Hua Wei[4], Ze Zhang[1✉], Frederic Sansoz [5✉] & Jiangwei Wang [1✉]

Nanoscale materials modified by crystal defects exhibit significantly different behaviours upon chemical reactions such as oxidation, catalysis, lithiation and epitaxial growth. However, unveiling the exact defect-controlled reaction dynamics (e.g. oxidation) at atomic scale remains a challenge for applications. Here, using in situ high-resolution transmission electron microscopy and first-principles calculations, we reveal the dynamics of a general site-selective oxidation behaviour in nanotwinned silver and palladium driven by individual stacking-faults and twin boundaries. The coherent planar defects crossing the surface exhibit the highest oxygen binding energies, leading to preferential nucleation of oxides at these intersections. Planar-fault mediated diffusion of oxygen atoms is shown to catalyse subsequent layer-by-layer inward oxide growth via atomic steps migrating on the oxide-metal interface. These findings provide an atomistic visualization of the complex reaction dynamics controlled by planar defects in metallic nanostructures, which could enable the modification of physiochemical performances in nanomaterials through defect engineering.

[1] Center of Electron Microscopy and State Key Laboratory of Silicon Materials, School of Materials Science and Engineering, Zhejiang University, Hangzhou 310027, China. [2] School of Mechanical and Electrical Engineering, Guilin University of Electronic Technology, Guilin 541004, China. [3] Institute of Molecular Plus, Tianjin University, Tianjin 300072, China. [4] Center for Hypergravity Experimental and Interdisciplinary Research, Zhejiang University, Hangzhou 310027, China. [5] Department of Mechanical Engineering and Materials Science Program, The University of Vermont, Burlington, VT 05405, USA. [6] These authors contributed equally: Qi Zhu, Zhiliang Pan, Zhiyu Zhao. ✉email: zezhang@zju.edu.cn; frederic.sansoz@uvm.edu; jiangwei_wang@zju.edu.cn

Nanotwinned materials possess a wide range of unprecedented mechanical and physical properties such as ideal maximum strength[1–3], excellent ductility[4], good fatigue resistance[5], high electrical conductivity[6], and exceptional thermal stability[7] due to the existence of coherent twin boundaries (TBs). With the advent of nanotechnology, TBs have also been revealed to critically influence the catalytic reactivity[8], electrochemical efficiency[9], growth dynamics[10], and oxidation rate[11,12] of metallic nanomaterials, which can be ascribed to the change of both surface properties (e.g., coordination number, stress state, and charge density) and internal lattice dynamics (e.g., diffusion) by TBs. Since the reaction dynamics at TBs play an important role in subsequent structural evolution and functionality of metallic nanomaterials[10,12,13], an atomic-scale understanding of this coherent defect contribution to chemical/electrochemical reactivity in nanomaterials is of general significance. However, in stark contrast to the well-established understanding of strengthening and softening mechanisms in nanotwinned materials[3,14,15], the atomistic origins of TB-assisted reactivity in nanotwinned materials have been largely elusive.

Recent studies of nanotwinned metallic nanoparticles attributed their superior catalytic performance to low-coordinated atomic steps, large tension, and high density of negative charges on the surface, which are critically affected by the presence of TBs[16–20]. These studies suggest that coherent TBs act as preferential sites for selective chemical/electrochemical reactions and as channels for fast atom transport[19,21]. Likewise, experimental studies have found that TBs significantly reduce the energy barriers for growth[13], mass transport[9], and solid-state reaction[22]. On the contrary, the twin-modified surface structures were reported to enhance the resistance to corrosion and oxidation in nanotwinned metals, compared to their twin-free nanocrystalline counterparts[12,23]. Therefore, the critical influence of TBs on the chemical response of metallic nanomaterials under different environments is still under debate. To date, however, systematic investigations of the defect-assisted oxidation dynamics at the atomic scale remain challenging in practice.

Here, we study the oxidation dynamics of nanotwinned metallic nanocrystals in atomistic detail by conducting integrated in situ transmission electron microscopy (TEM) characterizations and density-functional-theory (DFT)-based ab initio calculations. The results expose a unique site-selective oxidation behaviour in nanotwinned silver (Ag) and palladium (Pd), where the inherent planar defects including coherent TBs and stacking faults (SFs) strongly favour surface oxygen binding and accelerate oxide nucleation. Subsequent inward growth of the oxide is governed by the promoted oxygen diffusion along the planar defects that intersect with the oxide–metal interface, leading to layer-by-layer oxidation. Furthermore, we qualitatively validate this defect-assisted oxidation dynamics in air over largely different time scales spanning from few seconds to several days. These findings provide atomistic visualization and mechanistic understanding of defect-assisted reaction dynamics in nanoscale metals, which has direct ramifications for the development of advanced nanomaterials through defect engineering.

## Results

### Selective oxidation of nanotwinned Ag along isolated TBs and SFs.
In contrast to the widely reported oxidation of nanosized face-centred cubic (FCC) metals (e.g., Cu and Pd) from the {111} and {100} facets[24–26], our in situ experiments unambiguously demonstrate preferential oxide nucleation at the TB-surface junction, even in the presence of numerous low-energy (111) facets on the nearby surface. As shown in Fig. 1a, the as-fabricated Ag nanocrystal consists of a coherent TB along the axial direction, which is confirmed by the fast Fourier transform (FFT) pattern in Fig. 1f. Upon oxidation, sequential images clearly show that an oxide embryo preferentially nucleates at the twinned tip (Fig. 1b), although the lattice structure of this as-formed oxide embryo cannot be quantitatively identified. After 88 s, recognizable lattices appear as the oxide grows continuously inward along the TB (Fig. 1c). Meanwhile, the oxide expands laterally to form an asymmetrical conical cap sitting on top of the nanocrystal (Fig. 1d). No oxidation was detected on {111} facets away from the TB, suggesting a strong site-selective behaviour upon oxidation of this nanotwinned Ag. A close-up high-resolution TEM image is presented in Fig. 1e to show the atomistic structure of the oxidation product ($Ag_2O$) above the Ag matrix at the tip. The interface is atomically sharp with a one-atom-layer (111) step. The corresponding FFT patterns (Fig. 1g) further confirm the as-formed oxide of $Ag_2O$ with a lattice constant of $a$ = 4.718 Å (space group: $Pn\bar{3}m$), which bears a specific orientation relation with the Ag matrix as $(1\bar{1}0)_{Ag}//(100)_{Ag_2O}$ and $(11\bar{1})_{Ag}//(010)_{Ag_2O}$. It is further observed that oxidation along the TB always precedes that of the neighbouring lattices, as shown in Fig. 1e, which reflects an evident TB-assisted oxidation. Such selective oxidation mechanism is more conspicuous during the subsequent growth of this oxide layer, because the thickness of oxide (along axial direction) decreases gradually from ~ 3.4 nm at the tip down to almost zero at free surfaces on both sides (Fig. 1d). No other defect activity is observed throughout the oxidation process of this Ag nanocrystal.

Atomic-level snapshots in Fig. 1h–k further emphasize the preferential oxide nucleation at the TB-surface junction. A zoomed-in image shows the atomic structure of the Ag nanocrystal with a clean surface, which exposes two conjugate (111) facets with multiple atomic kink steps very close to the TB-surface junction, while a high density of alternate (111) and (11$\bar{1}$) atomic facets dominate on both sides of the surface away from the TB (Fig. 1h). The oxygen atoms are preferentially adsorbed to the surface layer of Ag near the TB (shown by the blurred profile of the top surface in Fig. 1i), which induces an evident lattice distortion in the sub-surface layers. In contrast, the faceted morphology of the surface away from the TB is generally retained, indicating less oxygen adsorption on these low-energy facets. Eventually, the adsorbed oxygen accumulating at the TB-surface junction incubates an oxide nucleus with an average height of 4.8 Å (Fig. 1j). With prolonged oxidation, the $Ag_2O$ layer simultaneously grows inward along the TB and expands laterally along conjugate oxide–metal interfaces (Fig. 1k).

The above experiment demonstrates a TB-assisted site-specific oxidation behaviour in Ag nanocrystal, which dominates over those widely reported low-energy surface facets for the nucleation and growth of oxides[24–26]. This unique behaviour is further validated in additional examples of Ag nanocrystals with an axial coherent TB (see Supplementary Fig. 1), where the TB-surface junction serves as a universally preferential nucleation site for oxide embryos, followed by oxide growth along the TB. Similar oxidation behaviours are also observed in other FCC metallic nanocrystals, e.g. Pd, with significantly higher stacking fault energy (Supplementary Figs. 2–4). This common site-selective nucleation and TB-assisted growth of oxide clearly indicate that an isolated TB can effectively promote the oxidation of FCC metals, which markedly differs from the suppressed chemical reactivity of nanotwinned Cu nanowires reported previously[27].

Furthermore, this phenomenon was found to occur with an individual TB positioned away from the Ag tip (i.e., with different surface curvatures, see Supplementary Fig. 5), which suggests that the TB effect is mainly intrinsic. DFT-based atomistic simulations

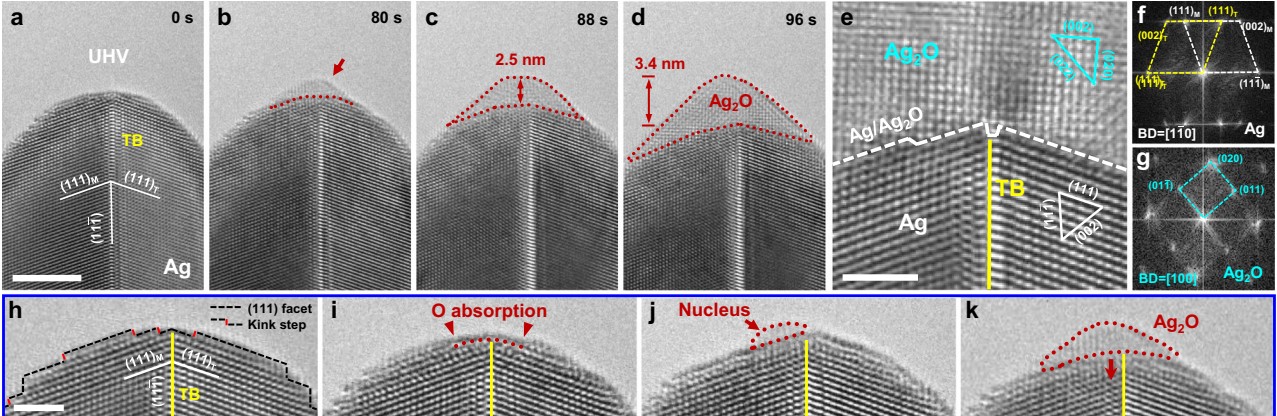

**Fig. 1 Twin boundary-favoured oxidation of Ag nanocrystal. a** Pristine Ag nanocrystal with a pre-existing twin boundary (TB) along the axial direction, which was placed in the ultrahigh vacuum (UHV) column of the transmission electron microscope (TEM). The nanocrystal has a clean surface with dominant (111) facets. **b–d** Dynamic snapshots showing the oxidation process of the Ag nanocrystal. The oxide embryo preferentially nucleated at the TB-surface junction (**b**), which grew continuously along the TB and expanded laterally on the nanocrystal surface (**c**, **d**). The red dotted loop delineates the oxidized region. **e** Zoomed-in image showing the atomistic structure of the as-formed $Ag_2O$ (top) and the Ag matrix (bottom). **f**, **g** Fast Fourier transform (FFT) patterns of the Ag nanocrystal with an axial TB and the oxidation product of $Ag_2O$, respectively. **h–k** Atomistic nucleation and growth of oxide at the surface–TB junction. **h** Clean surface of the nanocrystal with alternate (111) atomic facets and kink steps, where a higher density of kink steps pre-exist near the TB. **i** Preferential surface adsorption of oxygen at the TB tip, as indicated by the red dotted curve. **j** A tiny oxide nucleus formed at the TB-surface junction. **k** Continuous growth of $Ag_2O$ along the TB, as shown by the red arrow. Scale bars in **a** 5 nm; **e**, **h** 2 nm.

were performed to quantitatively confirm our hypothesis by calculating the Ag–O surface binding energy ($E_b$), which further decoupled the intrinsic influence of TB from other atomic surface steps. We first constructed a twinned Ag tip (consistent with our experiments) with a junction between a vertically aligned coherent TB and two perfectly smooth (111) conjugate surfaces. These calculations reveal that the O interstitial sites closest to the TB possess the highest mean oxygen binding energy $E_b = 3.907$ eV, compared with $E_b = 3.855$ eV at the neighbouring interstitial sites on the (111) surface farther away from TBs of the nanocrystal (Fig. 2a, b). Therefore, oxide nucleation should preferentially occur at the TB-surface junctions, which matches perfectly with our experimental observations. Furthermore, we find that the mean Ag–O binding energy at a kink step on the (111) surface ($E_b = 3.870$ eV, see Fig. 2c and Supplementary Fig. 6) is slightly higher than that at the smooth (111) surface, which may also serve as preferential oxygen absorption sites. These theoretical results are significant in three aspects. First, the DFT calculations quantitatively prove that both isolated TBs and kink steps give rise to stronger surface binding of O atoms. Second, it has been reported that twinning creates low-coordinated sites at the TB-surface junctions of nanotwinned metals, to which dioxygen preferentially bonds[28], suggesting that the reactivity of surface atoms maybe directly relate to their coordination number (CN). Here, our DFT-based calculations confirm that the higher oxgen binding energy is closely related to the lowest CN at a convex TB-surface (CN = 8), compared with its neighbouring sites (CN = 9). Third, we noted that the difference of mean oxygen binding energy between a twinned tip and a kink step without TB is comparatively small (Fig. 2b, c). However, it is important to emphasize that these two types of binding site structures are not directly comparable because it is only justifiable to study the two equivalent sites for the twinned tip and smooth (111) facets, as opposed to surveying multiple sites with severely fluctuant binding environments at the kink step (Supplementary Fig. 6). Occasionally, $E_b$ of certain binding sites at the kink step can even exceeds that at the TB tip. Nevertheless, the mean binding energy at an individual TB surpasses that at a kink step, despite the naturally lower CN = 7 of the kink step. Therefore, we can conclude that the lowest CN is a

good indicator for local maximum of $E_b$ when comparing the binding sites near or away from the TB in the same model, but this is not the case when comparing sites across completely different surface configurations.

**Oxidation efficiency as a function of defect configuration and spacing.** Analogous to the strong twin-size dependence of mechanical behaviours in nanotwinned metals[1–3], it is significant to study how the chemical reactivity and oxidation behaviour could vary with TB spacing, because smaller twins are frequently observed to reduce the reactivity via surface modification[12]. To this end, DFT calculations are used to quantify the mean oxygen binding energy averaged over multiple O interstitial sites of different defect-modified surface configurations (as shown in Fig. 2d and Supplementary Fig. 7), which overcome the technical difficulties in controlling TB spacing experimentally. Specifically, we calculated the mean binding energies of different nanotwins (NTs) with a constant TB spacing varied from 4.8 Å to 28.8 Å, in comparison to that of two single-crystal configurations containing either an isolated SF or no imperfection (SGL), as shown in Supplementary Figs. 8 and 9. The surface intersecting either TB or SF is composed of zig-zag low-energy (111) and (001) facets (insets of Fig. 2d). For consistency, the perfect single crystal configuration was constructed with the same type of zig-zag facets on the surface. Figure 2d shows that the mean $E_b$ of NTs is noticeably higher than that of SGL, which agrees with our experimental observations (Fig. 1). However, the binding energy tends to decrease as the TB spacing is reduced, which is in good agreement with the uniform and slower oxide growth observed in regions of copper nanowires containing dense and evenly spaced TBs[12,27]. The lower binding energy with decreased TB spacing could be attributed to the local change in the bonding environment of the oxidation site at the surface. Specifically, we have found that the oxidation sites with neighbouring atoms of lower CN tend to possess higher binding energies. Atoms at the convex tip and on the neighbouring (111) surfaces have a CN of 8 and 9, respectively, while those at the concave V-edge TB site have a larger CN of 10. Increasing the twin density decreases the distance between the V-edge site and the tip site. When the distance between the two types of sites is small (i.e., low TB spacing), the

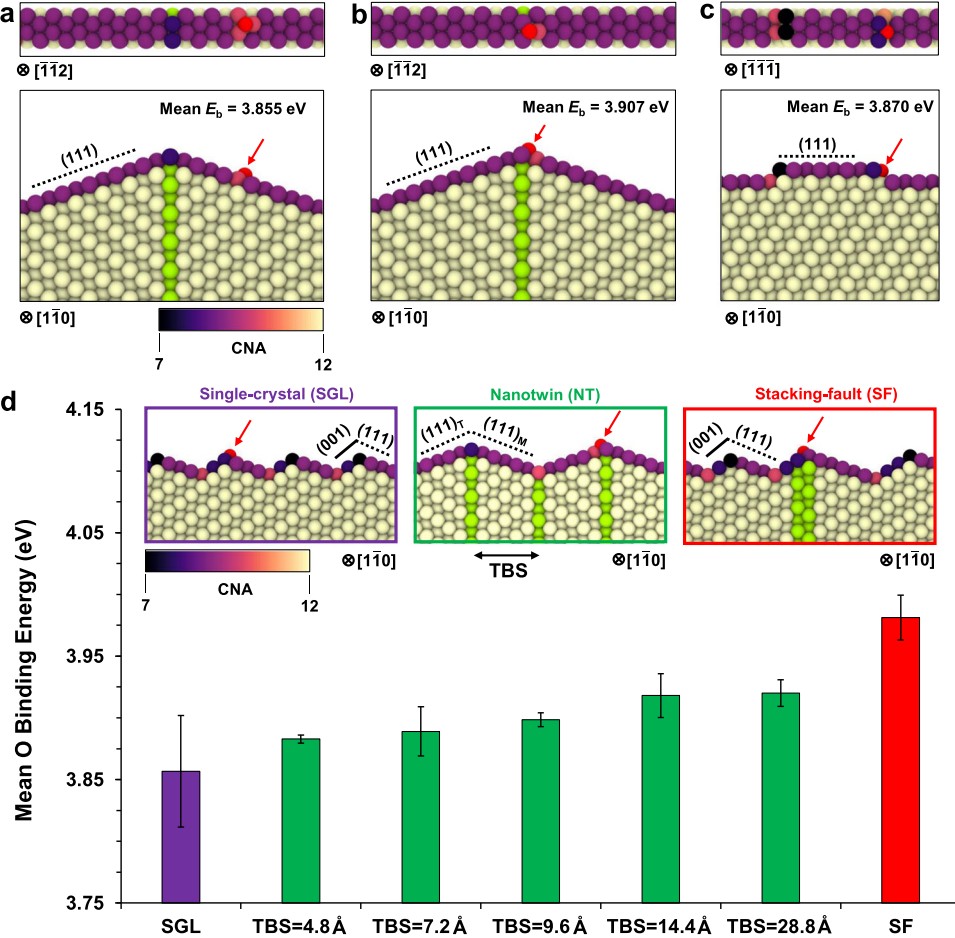

**Fig. 2 Density functional theory based calculations of the mean Ag–O binding energy at different surface sites of single crystal (SGL), nanotwinned (NT) and stacking fault (SF) contained Ag. a**, **b** Mean oxygen binding energy ($E_b$) at the atomically smooth (111) facets and the twinned tip, respectively. The oxygen atom is shown in red colour. Atoms at the planar faults are highlighted in green and other atoms in Ag are coloured using the coordination number analysis (CNA), as defined by the colour scale bar (black for lowest CN = 7 and yellow for highest CN = 12). **c** Mean $E_b$ for a (111) kink step in SGL Ag. **d** Mean $E_b$ at different surface sites of SGL, NT and SF contained Ag nanocrystals, where the twin-boundary spacing (TBS) varies from 4.8 Å to 28.8 Å. The highest mean $E_b$ is predicted to occur near isolated TBs (TBS = 14.4 Å and 28.8 Å) and near the SF. The different surface configurations are shown in the insets, where the (001) and (111) surface facets are delineated with solid and dotted lines, respectively. Error bars represent the standard deviations from statistical analyses, where the two highest energies (n = 2) are chosen for each nanocrystal configuration.

increased average CN among the neighbouring atoms at the tip will reduce the binding energy. Such effects on the average CN and bonding environment become substantial when the V-edge site is close enough to the tip, which explains the negligible change of $E_b$ from TBS = 28.8 Å to 14.4 Å followed by considerable drop with decreasing TBS thereafter. Even more remarkably, the mean $E_b$ at an isolated SF reaches 3.981 eV (Supplementary Fig. 9), which surpasses those of NTs with any size, indicating a preferential oxidation at the SF as compared with NTs. It is worth mentioning that the V-edge at the TB-surface junction (concave configuration) could act as a pinning site for diffusing Cu adatoms and promotes the dynamic formation of a Cu atom row (termed as W-chain) with lower CN and high chemical reactivity[28]. Therefore, while the CN principle is generally consistent with the results of our theoretical calculations, certain local differences may exist experimentally at the concave and convex configurations due to dynamic surface atom reconstruction.

Additional experiments in samples with different planar defects further confirm the strong site-selective and defect-assisted surface oxidation in Ag and Pd nanocrystals, as shown in Fig. 3 and Supplementary Figs. 4 and 5. Specifically, Fig. 3a–d

gives clear evidence for the SF-assisted oxidation in Ag nanocrystals with an isolated horizontal SF and multiple inclined TBs. The pre-existing SF laid almost perpendicular to the [111] axial direction of the Ag nanocrystal (Fig. 3a), intersecting the nanocrystal surfaces on both sides. A close-up image in the inset shows that the surface is oriented almost perfectly parallel to the (112) plane, albeit some (111) kink steps exist near the surface-SF junction. Upon initial oxidation, two oxide embryos nucleate simultaneously at the SF-surface junctions on both sides (Fig. 3b). This observation agrees well with the highest O binding energy at SF predicted by DFT. Moreover, the oxide embryo is followed by inward growth along the SF (Fig. 3c, d) similarly to our previous observations at individual TBs, where the average oxidation rate (along the SF) is calculated to be ~0.06 nm s$^{-1}$ (Supplementary Fig. 10). In contrast, the nucleation and growth of the oxide embryo at the TB occurred far later than that at the SF, confirming the preferential oxygen adsorption at the SF-surface junction predicted in our DFT calculations.

To further compare the oxygen binding energy at the SF and TB under the same orientation, in situ oxidation experiments were conducted on an Ag nanocrystal containing SF and multiple parallel TBs with uneven spacing, as shown in Fig. 3e–h. The

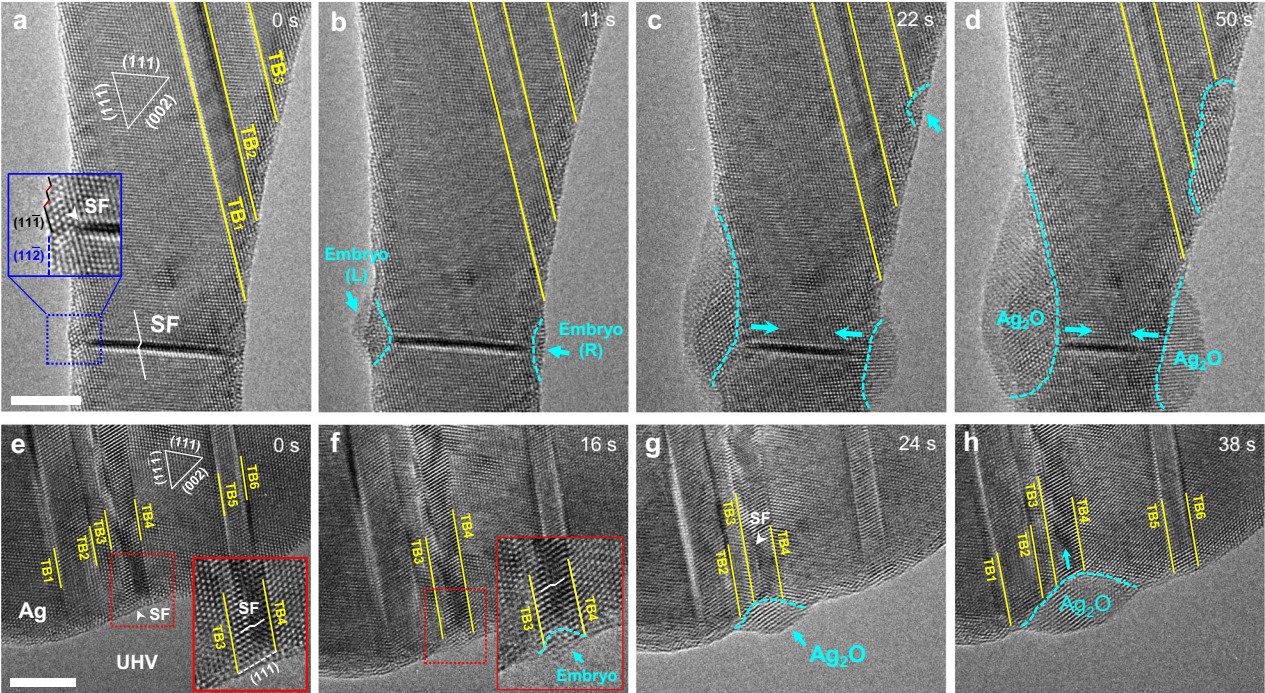

**Fig. 3 Selective oxidation of Ag nanocrystals with different planar defect configurations. a–d** Preferential oxidation of an Ag nanocrystal with an isolated SF. **a**, **b** Two oxide embryos formed preferentially at the SF-surface junctions on both sides (indicated by the light blue arrows), while the TB-surface junctions remained intact. Inset in **a** shows the atomistic structure of the surface near the SF, which consists of atomically flat (11$\bar{2}$) plane and several (11$\bar{1}$) kink steps. **c**, **d** Subsequent SF-assisted fast inward and lateral growth of $Ag_2O$ island. Meanwhile, oxide nucleation and growth was initiated at the $TB_3$-surface junction away from the SF. **e–h** Preferential oxidation in an Ag nanocrystal with unevenly spaced planar defects. **e** Multiple parallel TBs (#1–6) and a SF (indicted by the white arrow) pre-existed in the as-fabricated Ag nanocrystal with a clean surface. **f** Oxide embryo nucleated preferentially at the SF-surface junction. **g** Inward growth of the $Ag_2O$ island along the SF plane. **h** Integrated inward and lateral growth of the $Ag_2O$ oxide with the annihilation of the SF. No evident oxidation was observed at the TB-surface intersection sites nearby. Insets in **e**, **f** are zoomed-in snapshots presenting the atomistic nucleation processes of the oxide embryo at the SF-(111) surface facet junction. Scale bars in **a** and **e**: 5 nm.

surface configuration is almost identical near the intersection sites with either SF or TB (Fig. 3e), which decouples the influence of different surface facets on the defect-assisted oxidation. An oxide embryo preferentially nucleates at the SF-surface junction located between two TBs (TB3 and TB4 with a spacing of 1.88 nm), connecting a large twin to one of the smallest twins (Fig. 3f). It is worth noting, however, that no observable oxidation occurs at the nearby surface intersected by TBs. Subsequent inward growth of this oxide island preferentially proceeds along the SF (Fig. 3g, h), suggesting that SF is a faster pathway for oxygen diffusion. Associated with the inward oxide growth is the lateral expansion of the oxide along the surface facets (consistent with that at the surface–TB junction), which generates a spindle-shaped oxide. In summary, strong site-selection dictates the defect-assisted surface oxidation in metallic nanocrystals containing multiple TBs and SFs, and the general significance of coherent planar defects as fast pathways for oxygen diffusion is unambiguously demonstrated (see an additional example in Supplementary Fig. 11).

**Defect-assisted oxidation mechanism and kinetics**. Figure 4a–d presents a series of TEM snapshots showing the atom-level dynamics of TB-assisted oxide growth during the twinned tip experiment in Fig. 1. Upon nucleation, the interface between Ag nanocrystal and as-formed $Ag_2O$ contains six single-atom-layer steps (Fig. 4a). For the next second, the pre-existing steps migrate laterally along the $Ag/Ag_2O$ interface away from the TB, while the atoms at the TB tip react into $Ag_2O$, inducing a pair of new steps (7 and 8) in the sub-layer (Fig. 4b). This specific event suggests that interstitial O atoms are more rapidly diffused along the TB to bind with freshly exposed Ag atom columns, during which short-

cut diffusions along the TB plane is likely to occur[29]. In the following seconds, each newly formed step moves away on the $Ag/Ag_2O$ interface (Fig. 4c) and gives space for the formation of new $Ag_2O$ lattices into sub-layers initiated from the TB (Fig. 4d). This layer-by-layer oxide growth is similar to the growth of CuO nanowires assisted by (002) TBs[13], and eventually generates a conical oxide cap on the Ag nanocrystal (Fig. 1d).

The layer-by-layer growth mechanism is schematically illustrated in Fig. 4e, where the TB atoms in the top Ag layer and atomic steps on the $Ag/Ag_2O$ interface naturally possess a lower CN than those in the perfect FCC lattice (CN = 12). In fact, our DFT calculations have indicated that the CN of FCC Ag atoms decreases to CN = 10 at the $Ag/Ag_2O$ interface (Supplementary Fig. 12). Therefore, these low-coordinated Ag atoms at the oxide interface favour oxygen binding and induce a ledge flow mechanism at the reaction front[30], which is supported by experimental evidence (Fig. 4a–d). Furthermore, the surface oxide sitting on the nanotwinned tip is observed to grow continuously, suggesting that the coupled axial and lateral propagation of the $Ag/Ag_2O$ interface arises from fast diffusion of O atoms along the TB[31]. Similar TB-assisted diffusion has been reported in $SnO_2$ nanowire for Li ions (during lithiation)[9] and in α-Ti for O interstitials[32]. Also, the same evidence of TB-assisted layer-by-layer oxidation dynamics was found in our additional experiments of nanotwinned Pd, as exemplified in Supplementary Fig. 3.

The time-dependent oxidation kinetics along the TB is quantified in Fig. 4f. The TB-assisted oxidation of Ag nanocrystal (shown in Fig. 1) can be divided into three stages, i.e., fast nucleation, steady growth, and self-limiting equilibrium. During the fast nucleation ($t \leq 80$ s), an oxide embryo nucleates by

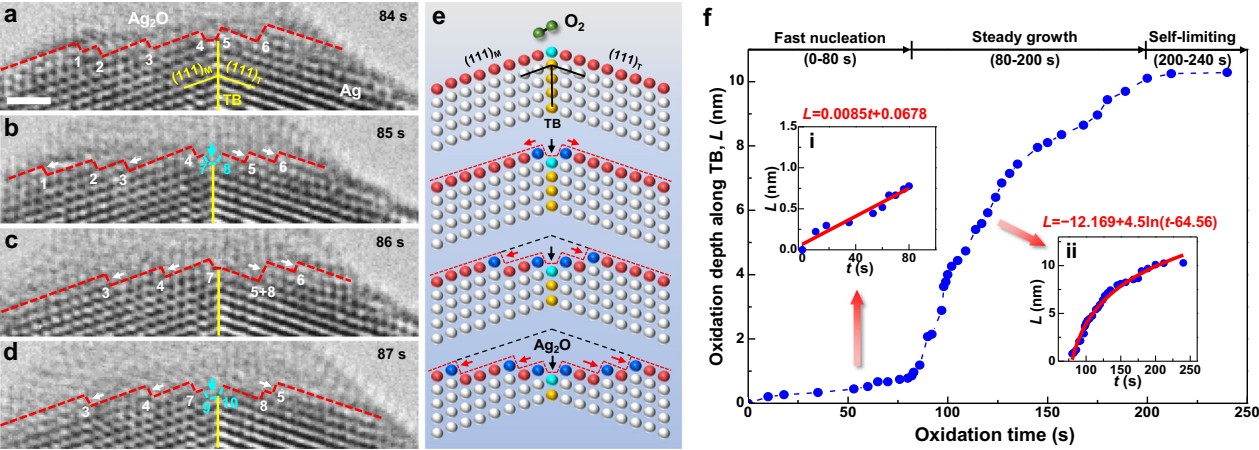

**Fig. 4 Dynamics of TB-assisted oxidation in Ag nanocrystal. a** The sharp reaction interface between the Ag matrix and the as-formed Ag₂O containing atomic steps (#1–6). **b** Inward oxidation along the TB, as shown by the dissolution of atoms at the TB tip (pointed out by the aqua arrow) into the oxide (Ag₂O), which induces two new atomic steps (#7 and 8). **c, d** Consecutive nucleation of interface steps from the TB junction, which migrate laterally along the conjugate (111) facets, leading to layer-by-layer oxidation of the Ag nanocrystal. **e** Schematic showing the layer-by-layer kinetics of TB-assisted oxidation in Ag nanocrystal. The atom columns at the TB tip, interface steps, and interface terraces are coloured aqua, dark blue and red, respectively. **f** The time-dependent inward oxide growth (characterized by oxidation depth, $L$) along TB in the Ag nanocrystal presented in Fig. 1. The oxidation depth-time plot was obtained by measuring the distance between the original surface–TB junction and the advancing interface–TB junction. Insets are the mathematical fitting of time-dependent oxide growth during different stages. Scale bar in **a**: 1 nm.

adsorption of rarefied oxygen atoms from the ultrahigh vacuum chamber of TEM. Our DFT calculations have predicted intrinsic TB and SF effects that yield a substantial increase in the surface binding energy of oxygen in Ag nanocrystals. Quantitative analysis in inset 1 of Fig. 4f confirms this theory by revealing that the axial oxidation depth $L$ (along TB) during the nucleation period is a linear function of time. A linear-type reaction is commonly associated with initial fast oxidation observed on a number of metals at low temperatures[33]. Subsequently, TB-assisted inward growth of the oxide proceeds layer-by-layer (Fig. 4e), which is referred to as the steady-growth stage (80 s < $t \leq$ 200 s). We have also plotted the cumulative step migration at the reaction front with time during the initial stage of inward oxide growth (Supplementary Fig. 13), which is consistent with the trend of $L-t$ plot, indicating the dominant role of layer-by-layer oxide growth via the TB-promoted nucleation and migration of interface steps. Our quantitative study in inset 2 of Fig. 4f further shows that the overall oxide growth kinetics follows a logarithmic law during steady growth, which is typically due to the electron tunnelling and ionic O flow in very thin oxide films[33]. However, the observation of spindle-shaped oxide islands (see Fig. 3 and Supplementary Fig. 2) indicates that oxygen diffusivity along TBs in the thin oxide layer is generally enhanced in comparison to the diffusivity on close-packed {111} planes in FCC metals, which agrees well with recent DFT calculations[32]. Meanwhile, the high oxygen concentration radiates from the TB to propel the lateral flow of newly formed steps along the Ag/ Ag₂O reaction front with low CN. As the oxide thickness grows to ~10 nm, however, few O ions are pumped towards the oxidation front (i.e., short of oxygen supply), leading to a gradually decreased oxide growth rate. This process is self-limiting under our experimental condition, because oxygen is rare in the TEM chamber. A similar oxidation trend is observed along an SF as well (see Supplementary Fig. 10).

## Discussion

It is established that by tunning the defect density and configurations, the surface morphology, strain gradient, or charge density can be changed markedly[16–19,27], enhancing the chemical reactivities in metallic nanoparticles. In essence, the TB-mediated

oxidations have already been reported[11]; however, the kinetic origin of the defect-facilitated oxidation at the atomic scale has been largely elusive. Here, our integrated in situ TEM observatrions and DFT calculations unambiguously elucidated a CN-governed site-selective oxidation at both the TB and SF in FCC metallic nanocrystals. The highest $E_b$ at the TB/SF–surface junctions favours the preferential oxygen adsorption and thus oxide nucleation; meanwhile, the enhanced oxygen diffusion along the planar defects greatly facilitates the continuous formation of steps at the reaction front and their lateral migration along the oxide–metal interface, leading to layer-by-layer growth of the oxide. Namely, the site-selective oxidation of Ag and Pd nanocrystals with pre-existing planar defects is not only controlled by the nucleation dynamics, but also facilitated by the diffusive growth kinetics, resulting in a nucleation-growth coupling of oxidation. This coordinated oxidation mechanism of metallic nanocrystals advances our mechanistic understanding of the defect-assisted reactivities.

From the thermodynamic perspective, the formation of oxide on the Ag surface is governed by the change of free energy, which can be derived according to Eq. (1):

$$\Delta G = \Sigma A_s \gamma_s + \Sigma A_i \gamma_i + \Delta G(Ag_2O) - \Sigma A_{Ag} \gamma_{Ag} \qquad (1)$$

where $\gamma_s$, $\gamma_i$, and $\gamma_{Ag}$ represent the surface energies of the oxide, oxide–metal interface, and the Ag matrix; $A_s$, $A_i$, and $A_{Ag}$ denote the corresponding surface or interface area; $\Delta G$ (Ag₂O) is the volume energy of Ag₂O[34]. It is widely acknowledged that the chemisorbed layer on the fresh metallic surface reduces the corresponding surface energy, resulting in a decreased value of $\Sigma A_{Ag} \gamma_{Ag}$. Therefore, the surface adsorption of oxygen atoms impairs the intrinsic dynamics for further oxidation. However, the higher oxygen binding energy at the TB/SF results in a site-selective oxygen adsorption, where only localized oxide embryo and oxide–metal interface forms, in stark contrast to the large area interface over the entire metal substrate. Therefore, the energy rise associated with the formation of oxide surface and oxide–metal interface can be significantly reduced, leading to a lower oxide nucleation barrier $\Delta G$. This qualitative thermodynamic analysis indicates that further oxidation process at room temperature should be essentially accelerated at the TB/SF, as

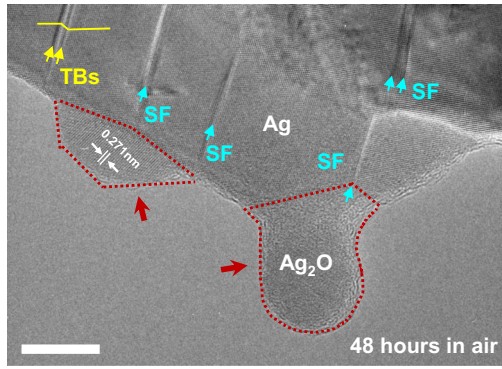

**Fig. 5 Natural oxidation of an Ag nanocrystal with multiple planar defects.** The Ag nanocrystal was exposed in ambient environment at room temperature for 2 days. The naturally oxidized Ag also exhibits similar site-selective oxidation behaviours along the pre-existing SFs. Scale bar: 10 nm.

verified with our experimental observations in Figs. 1 and 3. In contrast, homogeneous oxidation initiates from multiple sites simultaneously in defect-free nanocrystals, and the migration of the oxide–metal interface is mainly dominated by conventional bulk diffusion rather than the defect-assisted diffusion, leading to significantly slower inward growth (as shown in Supplementary Fig. 14 and schematically illustrated in Supplementary Fig. 15). Similar TB-assisted growth dynamics have also been reported in other systems such as CuO and Ge[13,35], providing general implications for the universality of the site-selective chemical reaction in a broad class of nanomaterials.

Despite the exceptional chemical inertness of bulk Ag and Pd, nanosized noble metals are inherently vulnerable to oxidation, which probably originates from the increased specific surface area. Under electron beam irradiation, the binding of oxygen with metallic atoms could be significantly facilitated at room temperature[36,37], even under low oxygen partial pressure in the ultrahigh vacuum TEM column ($\sim 2.5 \times 10^{-7}$ Torr). Owing to the high thermal conductivity of Ag (429 W mK$^{-1}$) and Pd (72 W mK$^{-1}$), temperature rise induced by the electron beam should be negligible in Ag and Pd nanocrystals[38]. However, the electron beam could also alter chemical bonds in the samples and induce ionization of gas molecules[39], thereby catalysing the oxidation. To exclude the possible beam effects on oxidation, nanotwinned Ag with multiple pre-existing TBs and SFs (fabricated with the same method) was exposed in air at room temperature for 48 h. The TEM image captured after natural oxidation (Fig. 5) shows that the naturally oxidized nanotwinned Ag underwent conspicuous site-selective oxidation along pre-existing isolated SFs, where negligible oxide was formed at defect-free areas nearby (Fig. 5), consistent with our in situ observation inside the TEM chamber. It needs to be clarified that although the size of the oxide appeared to be larger than those inside TEM, the oxide growth still appeared to proceed through a layer-by-layer mechanism, where the size of the oxide keeps increasing as long as the oxygen is stably supplied. Therefore, the extensive oxide growth in air (Fig. 5) should be reasonable, given the apparently higher oxygen pressure compared with that inside the TEM chamber. This natural oxidation experiment provides qualitative evidence for the generally retained site-selective oxidation mechanism under different oxygen concentrations and oxidation conditions, as well as to the time scale for oxidation.

In conclusion, the in situ atomic-scale observations and DFT calculations in this work have revealed a site-selective preferential oxidation behaviour in nanotwinned metals at planar defects including TBs and SFs. Such site-selective dynamics originates from the synergistic high binding energy at the TB/SF-surface

junctions and enhanced diffusion along TBs or SFs in metals, which is critically influenced by the planar defect density. Moreover, this defect-assisted surface oxidation mechanism underscores the general significance of the coupling between nucleation and subsequent growth dynamics upon chemical reactions in metallic nanocrystals, unravelling the relations between lattice defects and physicochemical phenomena at the atomic scale. Given that nanoscale twins and SFs are common in metallic nanocrystals with low stacking fault energies, this site-selective mechanism should play important roles during the nanowire growth, catalytic/chemical/electrochemical reactions, and solid-state phase transformation in a wide range of materials[16–19,40,41]. Therefore, our findings of atomistic reaction dynamics of defective nanocrystals hold fundamental and technological ramifications for the development of advanced nanomaterials through defect engineering.

## Methods

**In situ TEM oxidation**. In situ oxidation experiments were performed in a FEI Titan G$^2$ 60–300 transmission electron microscope equipped with a single-tilt STM-TEM holder from Zeptools Co. Before in situ oxidation experiments, Ag and Pd rods (0.25 mm in diameter, 99.99 wt% purity, ordered from Alfa Aesar Inc.) were mechanically fractured using a ProsKit wire cutter to obtain a clean fracture surface with numerous nanoscale tips. Due to the severe plastic deformation in the fracture zone, numerous nanoscale twins were induced in the nanotips on the fracture surface. Then, an Ag or Pd rod was loaded on to the TEM holder and inserted instantly into the TEM chamber. To characterize the twin structure, nanoscale tips in the <110> zone axis were selected for the in situ oxidation experiments. Then, in situ oxidation experiments were conducted at room temperature under ultrahigh vacuum (UHV) atmosphere $\sim 2.5 \times 10^{-7}$ Torr (corresponding to an oxygen partial pressure of $\sim 5 \times 10^{-8}$ Torr) to characterize the oxidation dynamics at the atomic scale. The extremely low oxygen pressure in the TEM chamber is sufficient to induce evident oxidation of the nanoscale metal. Besides, some oxygen may be pre-adsorbed on the fracture surface of Ag and Pd rods. For natural oxidation experiments, the fractured Ag rods were exposed in air at room temperature for 48 h before ex situ TEM characterization. In all experiments, the TEM was operated at 300 kV with a low electron beam current density of $\sim 100$ A s$^{-1}$ to minimize the possible beam effects. To investigate the oxidation dynamics, a Gatan 994 charge-coupled device (CCD) camera was used to record all experiments in real time at a rate of $\sim 0.3$ s per frame.

**First-principles calculations**. Ab initio calculations were performed using the Vienna ab initio simulation package (VASP) implemented based on the Kohn–Sham density functional theory[42]. Plane-wave basis set was used to expand the Kohn–Sham orbitals with an energy cutoff of 450 eV. The exchange-correlation functional was in the form of generalized gradient approximation developed by Perdew, Burke, and Ernzerhof (PBE)[43]. The projector augmented plane-wave method (PAW) pseudopotential was used to reduce the number of plane waves near the nucleus. Geometry optimization was performed to calculate the local minimum potential energy of each atomic configuration, where the ion positions were updated until the maximum force acting on the atoms was less than $1 \times 10^{-4}$ eV Å$^{-1}$. Since the simulation cell was large enough in X and Y direction, but only $\sim 6$ Å in Z direction, $k$-point calculation within a mesh of $1 \times 1 \times 3$ was performed for the wavefunction optimization with a convergence criterion of $1 \times 10^{-6}$ eV. The O binding energy, i.e., the energy required to remove one O atom from its bonded matrix configuration was calculated according to Eq. (2):

$$E_{\mathrm{b}}^{\mathrm{O}} = E_{\mathrm{ISO}}^{\mathrm{O}} + E_{\mathrm{Matrix}}^{\mathrm{Ag}} - E_{\mathrm{Matrix}}^{\mathrm{Ag, O}} \qquad (2)$$

where $E_{\mathrm{ISO}}^{\mathrm{O}}$, $E_{\mathrm{Matrix}}^{\mathrm{Ag}}$, $E_{\mathrm{Matrix}}^{\mathrm{Ag,O}}$ are the potential energy of an isolated O atom, that of the pure Ag configuration without O atom, and that of the same Ag configuration bonded with a surface O atom, respectively. For each Ag model, we studied between two and six different interstitial positions of surface O bond. For the single crystal, nanotwinned and SF models, the mean $E_{\mathrm{b}}$ was determined by averaging the two highest energies among neighbouring sites of a specific surface junction; for the kink step model, the mean $E_{\mathrm{b}}$ was determined by surveying five interstitial sites with fluctuant binding environments, see Supplementary Information.

## Data availability

The data that support the findings of this study are available from the corresponding author upon reasonable request.

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

## Acknowledgements

J.W., Z. Zhang, and H.W. acknowledge the support of Basic Science Center Program for Multiphase Evolution in Hypergravity of the National Natural Science Foundation of China (51988101). J.W. acknowledges the support of the National Natural Science Foundation of China (52071284, 51771172, and 51701179) and the Innovation Fund of the Zhejiang Kechuang New Materials Research Institute (ZKN-18-Z02). Z.P. and F.S. are grateful for the support from the US Department of Energy (DOE) (grants no. DE-SC0016270 and no. DE-SC0020054) and the supercomputer Cori of the National Energy Research Scientific Computing Centre, supported by DOE contract no. DE-AC02-05CH11231.

## Author contributions

J.W. and F.S. conceived and guided the research. J.W., Q.Z., Z. Zhao, G.C., and N.C. designed, conducted, and analysed the experiments. Z.P. and F.S. developed, performed and analysed the DFT simulations. L.L., H.W., and Z. Zhang helped with data analysis. Q.Z., J.W., Z.P., and F.S. prepared the manuscript. All authors contributed to the data discussion and manuscript revision.

## Competing interests

The authors declare no competing interests.
