## [Peer Review File · Nature Communications]

REVIEWER COMMENTS

Reviewer #1 (Remarks to the Author):

The authors reported the effects of coherent planar defects on the oxidation of Ag and Pd nanomaterials featured with (twins and stacking faults). They showed that the intersection areas of twin/stacking faults with the free surface are more favorable for the oxide nucleation and growth, rationalized by DFT calculations of higher oxygen binding energies at these defects. The use of in-situ environment electron microscopy observations and DFT modeling to delineate the atomic-scale mechanisms is impressive. The work is interesting in terms of addressing the important role of planar defects in the surface reactivity of metals. The manuscript was organized well to clearly convey the results. However, in compliance with the high standard criteria set for articles published by Nature Communications, the following concerns should be taken into careful consideration in revision:

1. **Originality:** The similar phenomenon and ideas were reported by Nishimoto et al in 2018 (Inorg. Chem. 2018, 57, 2908–2916), although the metal (Cu) studied is different from Ag and Pd in present manuscript. That paper already demonstrated that twinning defects in porous Cu can facilitate the nucleation and growth of Cu₂O and twinned Cu has a much faster oxidation rate in comparison to that without (or with much less) twinning defects. Therefore, the authors are suggested to clearly address the difference and any new improvement (including experimental method, in situ TEM, sample preparation, and new theory/modeling, and so on) of their work from the published one.
2. **Universality:** materials in serve are typically exposed to much higher high gas pressures compared to the extremely low oxygen pressure (e.g., 5x10⁻⁸ Torr) in the manuscript. It is unclear if the results can find applicability to practically relevant conditions of oxidation.
3. **Surface curvature effect:** there are a lot of studies showing the significant effect of surface curvature on the oxidation behavior. The intersection areas of the planar defects with the free surface have different curvatures from the neighboring surface areas. It is necessary to address whether the surface curvature plays any role in the observed oxide nucleation and growth.
4. From the Figure 1a to b, the Ag TB region gradually becomes much brighter in TEM image contrast. It needs to explain what causes such TEM image contrast changes. Does the oxide nucleation and growth involve any TB diffusion that depletes material from the bulk or TB? The authors state that TBs can be channels for fast atom transport (line 55). Does the oxide growth require any and short-circuit diffusion in the oxide layer or oxide/metal interface, like reported in a recent paper (Atomic-Scale Mechanism of Unidirectional Oxide Growth, Advanced Functional Materials 30 (4), 1906504 (2020)).
5. In figure 1h, the side surface (lower right corner) undergoes oxidation at step edges before the oxide nucleation at the TB tip. This contradicts with the main point in the manuscript of preferential nucleation of oxides at the TB/SF.
6. Fig. 1i shows some lattice distortion (or blurry lattice contrast) of atomic columns from the outer surface layer to about three atomic layers deeper, what causes such lattice contrast changes? It is proposed that the tip areas of TB and SF are the active sites for oxygen adsorption, based on DFT computed Ag-O bonding energies. However, Fig. 1i shows that the lattice distortion takes across the entire surface area, not only just in the area adjacent to the TB tip, how can this difference be reconciled between the experimental observations and the DFT modeling, any surface diffusion involved?

7. The in-situ HRTEM images show the inhomogeneous oxidation at the TB and SF sites. The mechanism for the significant oxide formation is not clarified. For instance, Figure 5 shows the growth of large oxide islands that are much bigger than the size of intersection lines of the planar defects with the outer surface. It is necessary to address how such extensive oxide growth occurs.
8. Lines 138-140: the sentence "Furthermore, we find that the Ag-O binding energy at a kink step on the (111) surface is considerably increased from $E_b = 3.897$ eV (Fig. 2c) to $E_b = 4.078$ eV in the vicinity of a planar SF (Fig. 2d)", however, the existence of planar SF does not show up in Fig 2d. Additional images in Supplementary Materials to display it may be useful to avoid the confusion.
9. The Ag-O bond energies at the tip of twin boundary and the flat (111) surface are compared. However, most of the flat surfaces shown in the TEM images have some stepped feature. The comparison between the TB and stepped surface seems instead of a flat surface seems more reasonable. In addition, the bonding energy difference is only 0.034 eV, which represents a tiny difference to clearly draw the conclusion.
10. The top-right corner region of the HRTEM images (Figure 3a-d) shows some growth and contrast changes over time, what was happening there and why?
11. Figure 3e-h display oxide nucleation at the SF and lateral oxide growth along the surface facets, what is the mechanism for the lateral oxide growth? However, some adjacent TB areas still do not oxidize, why?
12. EELS may be useful to further confirm the nature (such as Ag₂O or AgO) of the oxide in addition to HRTEM imaging.

Reviewer #2 (Remarks to the Author):

The paper reports an extensive study of defect-mediated oxidation behavior of Ag and Pd nanocrystals. The authors observed the preferential oxidation at the metal surface intercepted by coherent planar defects (twin boundary and stacking fault) using in-situ TEM. The authors claimed that the defect-mediated oxidation is attributed to the site-dependent binding energy between interstitial oxygen and exposed metal atoms at surface or metal/oxide interface according to DFT calculations. The defect-modulated chemical reaction behavior at nanoscale would be attractive to general audience. The experimental data presented were well written and organized with reasonable interpretation. It is a good piece of work and suitable for publication in Nature Communications after addressing the following issues.

1. In general, the kinetics of metal oxidation involves the inward diffusion of oxygen or the outward diffusion of metal ions, and the chemical reaction at metal/oxide interface or oxide surface. According to the TEM observation, the reaction between interstitial O and metal atoms is suggested to be the rate-determining step assuming that the diffusion of interstitial O in Ag₂O is sufficiently fast. Since the TB/SF-surface junction sites are more attractive to interstitial O (high binding energy), the oxide embryos tend to nucleate and grow at the junction sites. The argument seems reasonable. However, the statement "Fast diffusion of oxygen atoms along the planar-fault highways" may cause misleading impression for dynamics of Ag oxidation. If interstitial O atoms diffuse fast along the TBs or SFs, why not oxide embryos form inside the crystal?

2. The oxide growth proceeds with axial and lateral migration of atomic steps. But, the oxidation kinetics was simply evaluated based on the increased oxide thickness along the TB with time (Fig. 4). I would recommend to plot the migration of atomic steps with time based on the in-situ TEM observation, which may provide more detail information regarding the layer-by-layer oxide growth mode. Moreover, please comment on how the pre-existed atomic surface steps affect the oxide growth kinetics.
3. The binding energy calculations were performed for interstitial O atoms on the free surface with different planar defects. However, the reaction between oxygen and metal atoms mainly occur at the metal/oxide interface during the oxide growth stage. I am wondering whether the trend of site-dependent binding energies for the metal/oxide system is similar to that for the Ag crystal only (free surface).
4. Parallel TBs usually lead to a zig-zag surface structure in nanowires. The TB-surface junctions may have “convex” and “concave” configurations. The DFT calculation presented in this study mainly address the convex configuration. I am curious whether the concave junction has the similar binding energy as the convex one. If not, it may have different oxidation kinetics.
5. The mean Ag-O binding energies in Fig. 2 were calculated at different surface sites of nanotwinned Ag with varied TB spacing. How many surface site data were averaged for each TB spacing case? Besides, some atomic steps may exist on the nanotwinned crystal surface, which could also depend on the TB spacing as reported in Ref. 12. How would the pre-existed atomic steps affect the calculations?

Reviewer #3 (Remarks to the Author):

In the study by Zhu et. al., authors investigate the mechanisms and kinetics of selective defect driven oxidation of nanocrystalline metals using in-situ TEM characterization and DFT computations. The results of the study are novel and of general interest to the materials science community. The conclusions are based on strong, technically sound evidence. I recommend this manuscript for publication if authors can answer the following questions:

- 1) What is the source of the large error-bars in Fig. 2e, especially, in the case of SGL and certain spacings of TBS? While the mean binding energy is higher for TBS, and further increases with increasing spacing, does this trend still hold for the range of binding energies computed for each case, given the large error bars in some cases?
- 2) While the authors have sufficiently discussed the mechanisms and kinetics of defect assisted nucleation and growth, can they comment on the thermodynamics of nucleation? Especially how does binding energy affects critical radius of nuclei that can spontaneously grow?
- 3) What is the physical reason behind higher binding energy of interstitial sites closer to twin/surface junctions and kink steps/sites near SFs or SFs/surface junction than sites/kink steps away from SFs and twins? Is it related to CN? However, I think authors state that “that the coordinate number (CN) of atoms at the TB-surface junction (CN=8) is larger than that at the edge of a (111) kink step (CN=7).”

4) Why does binding energy decrease with increasing twin density?

5) Authors state that “In contrast to nanotwinned Ag, however, we found that homogeneous oxidation initiates from multiple sites simultaneously in defect-free Ag single crystals (Supplementary Fig. 13); such process is mainly dominated by surface diffusion rather than defect-assisted diffusion, leading to significantly slower kinetics and reduced inward growth.” If the oxidation in pure Ag is dominated by surface diffusion rather than defect-assisted diffusion, shouldn't oxidation kinetics be faster in pure case, since surface has the highest diffusivity than most defects? Authors should be careful while making such statements.

Point-to-point Response to the Reviewers' comments

Manuscript ID: NCOMMS-20-25975

Title: Defect-driven Selective Metal Oxidation at Atomic Scale

Authors: Qi Zhu, Zhiliang Pan, Zhiyu Zhao, Guang Cao, Langli Luo, Chaolun Ni, Hua Wei, Ze Zhang, Frederic Sansoz, Jiangwei Wang

We sincerely thank the editor and the reviewers for their time and efforts in carefully reading our manuscript and providing valuable comments and constructive suggestions for us to improve the quality of this manuscript. In the following, the reviewers' comments are laid out in *italicized font* and our response to each comment is given in blue text. The manuscript has been revised accordingly and the changes have been highlighted in a red color.

Response to Reviewer #1

The authors reported the effects of coherent planar defects on the oxidation of Ag and Pd nanomaterials featured with (twins and stacking faults). They showed that the intersection areas of twin/stacking faults with the free surface are more favorable for the oxide nucleation and growth, rationalized by DFT calculations of higher oxygen binding energies at these defects. The use of in-situ environment electron microscopy observations and DFT modeling to delineate the atomic-scale mechanisms is impressive. The work is interesting in terms of addressing the important role of planar defects in the surface reactivity of metals. The manuscript was organized well to clearly convey the results. However, in compliance with the high standard criteria set for articles published by Nature Communications, the following concerns should be taken into careful consideration in revision:

Response: We sincerely appreciate the reviewer for recognizing the scientific importance of our work. We hope our responses below demonstrate that we have made necessary changes to address all of the concerns.

1. Originality: The similar phenomenon and ideas were reported by Nishimoto et al in 2018 (Inorg. Chem. 2018, 57, 2908–2916), although the metal (Cu) studied is different from Ag and Pd in present manuscript. That paper already demonstrated that twinning defects in porous Cu can facilitate the nucleation and growth of Cu₂O and twinned Cu has a much faster oxidation rate in comparison to that without (or with much less) twinning defects. Therefore, the authors are suggested to clearly address the difference and any new improvement (including experimental method, in situ TEM, sample preparation, and new theory/modeling, and so on) of their work from the published one.

Response: We appreciate the reviewer for the constructive suggestions with regard to the past study of Nishimoto et al. in 2018. We want to emphasize, however, that our work is an important step forward from these initial observations because our experimental and theoretical results at atomic scale provide fundamentally new understanding in several aspects.

First, in the work of Nishimoto *et al.* (Inorg. Chem. 2018, 57, 2908–2916), higher oxidation rates of nanotwinned Cu were quantitatively measured macroscopically with XRD and H₂-DSC, but lacked atomistic details on the step-by-step oxidation dynamics and kinetics at the TBs, especially in experiments. In our work, we successfully attained a direct visualization of the site-selective oxidation in nanotwinned Ag and Pd and revealed the **kinetic origin** at the atomic scale. Specifically, the atomistic kinetics of site selective oxide nucleation at TB (due to the higher oxygen binding energy at TB) and its layer-by-layer growth (facilitated by fast diffusion along the TB) have been thoroughly elucidated.

Second, an interlink between oxide nucleation and growth was proposed based on our *in situ* observations and DFT calculations. One of the novelties of our study is to systematically address the roles of TB spacing and surface structure on surface oxide nucleation, using DFT calculations, which were overlooked and thus unknown in these past studies.

Third, another new finding is that stacking faults can impose similar (or even more significant) influences on the oxidation of Ag and Pd, indicating a broader concept of site selective oxidation governed by planar defects (not limited to TBs) in metallic crystals. Previous literature reports have observed the TB-assisted nanowire growth among different material systems such as CuO and Ge nanowires (Rackauskas *et al.*, *Nano Lett.*, 2014, **14**, 5810-5813; Gamalski *et al.*, *Nano Lett.*, 2014, **14**, 1288-1292), thereby providing general implications for the universality of the site-selective chemical reaction in a broad class of nanomaterials.

We have highlighted these originalities in our revised manuscript by clearly presenting the new improvements of our work in Abstract and supplementing a brief comparison with previous work in Discussion, following the reviewer's suggestion.

2. Universality: materials in service are typically exposed to much higher high gas pressures compared to the extremely low oxygen pressure (e.g., 5x10⁻⁸ Torr) in the manuscript. It is unclear if the results can find applicability to practically relevant conditions of oxidation.

Response: We thank the reviewer for raising this important point. We are aware of the potential differences between the extremely low oxygen pressure in TEM chamber and the real service conditions. In fact, we have presented similar TB-assisted oxidation of Ag exposed to air at room temperature for 48 hours (see Fig. 5). Apparently, multiple extensive oxidations occurred at different planar defect-surface junctions, consistent with our observations inside TEM. This observation strongly suggests that the oxygen pressure has no significant impact on the defect-induced oxidation behavior. One may note that, the higher oxygen pressure should accelerate oxygen absorption over the entire surface and oxygen transport via bulk diffusion, which may challenge the site-selective kinetics. However, the observation of retained site-selective oxidation in ambient air proves the direct practical implication for the defect-facilitated oxide nucleation and growth mechanism proposed in our work.

3. *Surface curvature effect: there are a lot of studies showing the significant effect of surface curvature on the oxidation behavior. The intersection areas of the planar defects with the free surface have different curvatures from the neighboring surface areas. It is necessary to address whether the surface curvature plays any role in the observed oxide nucleation and growth.*

Response: We agree with the reviewer that the surface curvature can impose effects on the oxidation behavior. First, it needs to be clarified that the surface curvature can be classified into intrinsic curvature (original morphology of the surface) and the extrinsic curvature (due to the intersection between planar defects and surface). The intrinsic curvature imposes no observable influence on the TB-mediated oxidation, since the TB-surface junctions at different curvatures can induce similar site-selective oxidation behavior (as exemplified by the comparison between Fig. 1 and Supplementary Figs. 2, 4-5).

The physical origin of the extrinsic curvature effects is that the planar defect intersecting the surface naturally generates an either concave or convex local configuration (Nishimoto *et al.*, *Inorg. Chem.* **57** (2018); Rackauskas *et al.*, *Nano Lett.* **14** (2014)), where the metallic adatoms at the tips generally possess lower coordination number (CN) and higher chemical reactivity (Krajčí *et al.*, *J. Chem. Phys.* **145**, 084703 (2016)). This explanation qualitatively applies to our results as well, as shown in the DFT calculation in Fig. 2, where atom columns at the TB tips possess lower CN and higher oxygen binding energy. Besides, the surface curvature is also dependent on the spacing between planar defects, which has been elucidated in both DFT calculations (Fig. 2d and Supplementary Fig. 8) and *in situ* experimental observations (Fig. 3e-h). **In conclusion, we believe that the physical origins of defect-induced curvature effects are CN and defect spacing**, which we have further clarified in the revised manuscript.

4. *From the Figure 1a to b, the Ag TB region gradually becomes much brighter in TEM image contrast. It needs to explain what causes such TEM image contrast changes. Does the oxide nucleation and growth involve any TB diffusion that depletes material from the bulk or TB? The authors state that TBs can be channels for fast atom transport (line 55). Does the oxide growth require any and short-circuit diffusion in the oxide layer or oxide/metal interface, like reported in a recent paper (Atomic-Scale Mechanism of Unidirectional Oxide Growth, *Advanced Functional Materials* 30 (4), 1906504 (2020)).*

Response: We thank the reviewer for these insightful comments. The TB structure captured in Fig. 1a is right on focus. The brighter contrast of the TB in Fig. 1b corresponds to the defocused imaging of TB, due to the inevitable drift of such small and thin samples in the TEM chamber during the *in situ* experiment. Moreover, no clear evidence indicates the depletion of Ag via TB diffusion. Instead, the diffusive transport of oxygen atoms is promoted along the TB (Hooshmand *et al.* *Acta Mater.* **156** (2018)), facilitating the continuous inward oxide growth.

In term of the short-circuit diffusion mentioned by the reviewer, the layer-by-layer growth along the oxide-metal interface is qualitatively similar to the unidirectional growth of CuO along the (002) GB (*Adv. Funct. Mater.* **30**, 1906504 (2020)), where the oxide growth direction complies with the short-circuit diffusion via cation-vacancy exchanges. Therefore, we believe

that such short-circuit diffusion along the oxide-metal interface is likely to occur at the interface steps, which contribute to the step motion along the oxide-metal interface and the consequent layer-by-layer growth of the oxide. We have cited this work mentioned by the reviewer as Ref. 29 in the revised manuscript.

5. In figure 1h, the side surface (lower right corner) undergoes oxidation at step edges before the oxide nucleation at the TB tip. This contradicts with the main point in the manuscript of preferential nucleation of oxides at the TB/SF.

Response: We thank the reviewer for pointing out this problem. It needs to be clarified that the conspicuous contrast at the lower right corner arises from the residual surface contamination or carbon deposition induced by the electron beam shower before the *in situ* experiment (this is likely to occur during the sample inspection at the neighboring places). As shown in Fig. R1a, the Ag nanocrystal (sample in Figure 1 of the main text) is vulnerable to surface contamination/deposition inside the TEM chamber, especially under a weak electron beam (Zheng et al. *Nanoscale* 6, 9574-9578 (2014)). Therefore, we further performed a beam shower on the nanocrystal with higher dose rate to sublimate those contaminants, as shown in Fig. R1b. Since the nanocrystal tip was closer to the beam center, a clean surface was obtained after the strong beam shower, as shown in Fig. R1c (*i.e.* Fig. 1a and 1h of the main text); in contrast, the contaminants on the side surfaces away from the beam center underwent incompletely sublimation, and certain residuals were left.

To avoid any confusion, we have further clarified the surface conditions in the revised manuscript and update the TEM image accordingly.

Fig. R1. (a) Surface contamination or carbon deposition on the as-received twinned Ag nanocrystal. (b) Beam shower under four times the dose rate. (c) Clean surface at the nanocrystal tip after beam shower.

6. Fig. 1i shows some lattice distortion (or blurry lattice contrast) of atomic columns from the outer surface layer to about three atomic layers deeper, what causes such lattice contrast changes? It is proposed that the tip areas of TB and SF are the active sites for oxygen adsorption, based on DFT computed Ag-O bonding energies. However, Fig. 1i shows that the lattice

distortion takes across the entire surface area, not only just in the area adjacent to the TB tip, how can this difference be reconciled between the experimental observations and the DFT modeling, any surface diffusion involved?

Response: We thank the reviewer for these insightful comments. We are aware of the blurry lattice contrast due to the lattice distortion. Such lattice distortion (about three atomic layers) is attributed to the O atom entering the interstitial sites in Ag. Theoretically, the conspicuous change of lattice parameter from 4.08 Å (pure Ag) to 4.718 Å (Ag₂O) should cause localized lattice mismatch between the surface atom layer (after the formation of Ag-O bonds) and those underneath the surface, which is consistent with our observation of distorted lattice. First, we have double-checked the area of surface distortion and found that the red dotted lines sketched on our HRTEM image could be misleading. Therefore, we have removed some unnecessary labels and lines to better reveal the true extent of the blurry lattice contrast due to oxygen adsorption in the revised manuscript (see Fig. R2). Second, in the revised figure, the oxygen adsorption appears to occur throughout the top surface at both the TB tip and neighbouring kink steps (rather than involving substantial surface diffusion), indicating that both TBs and kink steps are active oxidation sites. Although it is difficult to experimentally separate the role of TB-surface junction and the kink steps since the oxygen was quickly absorbed over the top surface, the lattice distortion went deeper into the Ag nanocrystal near the TB (as shown at the intersection between the yellow and red lines in Fig. 1i), supporting the claim that the TB tip is a preferential site. To completely decouple the effect of kinks, we performed DFT calculations to study the ideal configurations with and without surface kink steps (as shown in Fig. 2), which confirmed the respective intrinsic roles of TBs and SF in increasing the O binding energy at the surface.

Fig. R2. Oxygen absorption at the TB-surface junction. The surface area within the red arrows underwent most conspicuous lattice distortion.

7. *The in-situ HRTEM images show the inhomogeneous oxidation at the TB and SF sites. The mechanism for the significant oxide formation is not clarified. For instance, Figure 5 shows the growth of large oxide islands that are much bigger than the size of intersection lines of the planar defects with the outer surface. It is necessary to address how such extensive oxide growth occurs.*

Response: A detailed discussion on the defect-assisted oxidation mechanism, supported by kinetics measurements, has been provided in the last section of the manuscript; however, we thank the reviewer for asking to further clarify this mechanism with respect to Fig. 5. After the site-selective oxide nucleation, subsequent oxide growth proceeds through a layer-by-layer

mechanism (or equivalently, step-flow mechanism). Specifically, the continuous oxygen diffusion along the TB and SF from the surface into the metal facilitates the nucleation of new steps underneath the oxide-metal interface, resulting in the inward growth of the oxide; the as-formed interface steps moved along the interface, leading to lateral expansion of the oxide. These two coordinated step evolution mechanisms contribute to the layer-by-layer oxidation, in which the size of the oxide keeps increasing as long as the oxygen is stably supplied. Therefore, the extensive oxide growth in air (Fig. 5) should be reasonable, given the apparently higher oxygen pressure compared with that inside the TEM chamber. We have added the above clarification to the analysis of Fig. 5 in the revised manuscript.

8. Lines 138-140: the sentence “Furthermore, we find that the Ag-O binding energy at a kink step on the (111) surface is considerably increased from $E_b = 3.897$ eV (Fig. 2c) to $E_b = 4.078$ eV in the vicinity of a planar SF (Fig. 2d)”, however, the existence of planar SF does not show up in Fig 2d. Additional images in Supplementary Materials to display it may be useful to avoid the confusion.

Response: In our atomistic model in Fig. 2d, a horizontal faulted plane parallel to the (111) surface was visible between the two kink steps, as highlighted by a row of green-colored HCP atoms. This single faulted layer was different in appearance from the intrinsic two-layer stacking fault shown in Fig. 2e, because we initially wanted to compare the exact same (111) surface structure with kinks between Figs. 2c and 2d. However, since the faulted kink step did not match any structure in our experiment, we have removed Fig. 2d and the associated discussion to avoid any confusion, as shown in Fig. R3. Furthermore, Supplementary Fig. 9 has been replaced with the model of single-crystalline kink step (fault-free), as shown in Fig. R4.

Fig. R3. Density-functional-theory calculations of the Ag-O binding energy at different surface sites of single-crystal (SGL), nanotwinned (NT), and stacking-faulted (SF) Ag. (a-b) Mean interstitial O surface binding energy E_b for sites on (a) an atomically smooth {111} facet, (b) twinned tip and (c) atomic (111) surface kink step. The

oxygen atom is shown in red colour. Atoms of TBs are highlighted in green and other atoms in Ag are coloured using the coordination number analysis (CNA).

Fig. R4. Site-specific O binding energies calculated by *ab initio* simulation in Ag single crystal with an atomic (111) surface kink step.

9. The Ag-O bond energies at the tip of twin boundary and the flat (111) surface are compared. However, most of the flat surfaces shown in the TEM images have some stepped feature. The comparison between the TB and stepped surface seems instead of a flat surface seems more reasonable. In addition, the bonding energy difference is only 0.034 eV, which represents a tiny difference to clearly draw the conclusion.

Response: We thank the reviewer for these constructive comments. In fact, we have considered the effects of stepped surfaces in DFT calculations. As shown in the revised Fig. 2 (Fig. R4 of the response) and Supplementary Fig. 6-9, the surfaces in the DFT calculations were intentionally constructed with low energy {111} steps, which are identical with our experiments. We have clearly stated this surface configuration in the revised manuscript according to the reviewer's suggestion.

From the DFT standpoint in Fig. 2b and 2c, we agree with the reviewer that the difference of O binding energy between a twinned tip and a kink step without TB is comparatively small. However, it is important to emphasize that the binding site structures are not so easily comparable because it is required to study only two sites for the twinned tip and smooth (111) facets, which are equivalent (Supplementary Fig. 8e), as opposed to survey multiple sites with severely fluctuant binding environments at the kink step (Supplementary Fig. 9). Therefore, we have revised Fig. 2a-c by showing the mean binding energy at an individual TB surpasses that at a kink step, despite the naturally lower CN (=7) of the kink step.

Furthermore, we have discussed in the manuscript that site-selective oxidation is controlled by both nucleation and growth kinetics. The binding energy governs the potential for the nucleation of oxide embryo, while the fast diffusion along planar defects guarantee the subsequent oxide growth. Therefore, both TB/SF and the nearest steps can facilitate successive oxide nucleation and growth, owing to the readily promoted oxygen diffusion, while those away from the planar defects play negligible roles in oxidation, as demonstrated in Fig. 1.

10. *The top-right corner region of the HRTEM images (Figure 3a-d) shows some growth and contrast changes over time, what was happening there and why?*

Response: The growth and contrast change at the top-right corner were also induced by the TB-mediated oxidation (see Fig. R5), which was only partly shown in Fig. 3. In Fig. R5, it is clear that the extent of oxidation at the TB-surface junction is not comparable to that at the SF-surface junctions, consistent with our DFT prediction. We have thus updated Fig. 3a-d in the revised manuscript.

Fig. R5. Site-selective oxidation at TB and SF in an Ag nanocrystal.

11. *Figure 3e-h display oxide nucleation at the SF and lateral oxide growth along the surface facets, what is the mechanism for the lateral oxide growth? However, some adjacent TB areas still do not oxidize, why?*

Response: We thank the reviewer for raising this important question. The mechanism of the lateral oxide growth at the SF should be similar to that at the TB tip, *i.e.*, inward layer-by-layer oxidation. The inward and lateral growth of the oxide are essentially coupled, in which the nucleation of a new step marked the initiation of inward oxide growth for one atom layer and the concomitant motion of steps along the oxide-metal interface led to the lateral expansion of the oxide. This mechanism is confirmed by the spindle-shaped oxide shown in Fig. 3h, which is identical to that shown in Fig. 1 and Fig. 3a-d.

Besides, we would like to emphasize that the competition between different sites (SF and TBs) determines the final oxidation process. The higher oxygen binding energy at the SF, as suggested by our DFT calculations, means that the sites at SF is more likely to start the oxidation and/or takes a shorter time for the oxidation to occur. Once the oxidation nucleates at these sites, subsequent oxide growth will draw O atoms nearby and create an oxygen depleted area that further reduces the chance of oxidation at nearby TBs.

12. *EELS may be useful to further confirm the nature (such as Ag_2O or AgO) of the oxide in addition to HRTEM imaging.*

Response: We agree that EELS is a useful technique to reveal the nature of the oxide (in addition to HRTEM imaging). However, due to the complicated setting up/calibration, EELS usually takes longer time to obtain the results, which is not appropriate to integrate with the *in situ* experiment to capture the fast-changing details at the oxidation site. Moreover, extremely limited signals from such small amount of oxide may not generate distinguishable peaks in the EELS spectrum and the strong beam current under STEM mode for EELS may even cause the reduction of Ag_2O . Besides, the obvious crystallographic differences between Ag_2O ($Pn3m$) and AgO ($P2_1/c$) can be unambiguously distinguished from HRTEM images by measuring the interplanar spacing and dihedral angles. Yet, we would like to explore any efficient combination of *in situ* observation and EELS characterization in the future.

Response to Reviewer #2

The paper reports an extensive study of defect-mediated oxidation behavior of Ag and Pd nanocrystals. The authors observed the preferential oxidation at the metal surface intercepted by coherent planar defects (twin boundary and stacking fault) using in-situ TEM. The authors claimed that the defect-mediated oxidation is attributed to the site-dependent binding energy between interstitial oxygen and exposed metal atoms at surface or metal/oxide interface according to DFT calculations. The defect-modulated chemical reaction behavior at nanoscale would be attractive to general audience. The experimental data presented were well written and organized with reasonable interpretation. It is a good piece of work and suitable for publication in Nature Communications after addressing the following issues.

Response: We sincerely thank the reviewer for recognizing the significance of our work and have done our best to address all the issues.

1. *In general, the kinetics of metal oxidation involves the inward diffusion of oxygen or the outward diffusion of metal ions, and the chemical reaction at metal/oxide interface or oxide surface. According to the TEM observation, the reaction between interstitial O and metal atoms is suggested to be the rate-determining step assuming that the diffusion of interstitial O in Ag₂O is sufficiently fast. Since the TB/SF-surface junction sites are more attractive to interstitial O (high binding energy), the oxide embryos tend to nucleate and grow at the junction sites. The argument seems reasonable. However, the statement “Fast diffusion of oxygen atoms along the planar-fault highways” may cause misleading impression for dynamics of Ag oxidation. If interstitial O atoms diffuse fast along the TBs or SFs, why not oxide embryos form inside the crystal?*

Response: We thank the reviewer for pointing out this instructive question. The statement “fast O diffusion” is based on the comparison between diffusion along SF/TB and the conventional bulk diffusion. This was validated by our experimental observation of steps on the oxide-metal interface that preferentially nucleated from the TBs. Despite this fast diffusion pathway along SF/TB, an oxygen gradient exists during the oxidation, with the maximum oxygen concentration at the surface. Therefore, oxide nucleation from the crystal interior is not favored. To avoid any misleading impression, we have changed this statement to “Planar-fault mediated diffusion of oxygen atoms” in the abstract of the revised manuscript.

2. *The oxide growth proceeds with axial and lateral migration of atomic steps. But, the oxidation kinetics was simply evaluated based on the increased oxide thickness along the TB with time (Fig. 4). I would recommend to plot the migration of atomic steps with time based on the in-situ TEM observation, which may provide more detail information regarding the layer-by-layer oxide growth mode. Moreover, please comment on how the pre-existed atomic surface steps affect the oxide growth kinetics.*

Response: We thank the reviewer for this constructive suggestion. We have plotted the migration of atomic steps with time as recommended by the reviewer, as shown in Fig. R6. Specifically, we recorded the cumulative cases of step migration N with time during the initial stage of oxide growth, in which the oxide lattice and all interface steps can be clearly resolved and tracked (the track-and-count criterion for each step motion is schematically illustrated in inset i). The evident rise of step migration rate is consistent with the accelerated increase of oxide thickness (L) during the same period of time, as shown by the inset $L-t$ curve (Fig. 4f in the manuscript), which further verifies the dominant layer-by-layer oxide growth via the consecutive step nucleation and migration. We have therefore added this plot as Supplementary Figure 13.

Fig. R6. Cumulative atomic step migration along with increasing oxidation time ($N-t$ curve) during the initial stage of oxide growth. The inset i schematically illustrates the track-and-count criterion for each step motion and inset ii is the oxide thickness-time ($L-t$) plot in Fig. 4 of the manuscript, where the red square marks the initial stage of oxide growth.

For the effects of pre-existed atomic surface steps, our DFT calculations demonstrated that the binding energy at kink steps are conspicuously higher than that at the flat surface (or even the TB-surface junction), potentially facilitating the oxide nucleation. However, as we have discussed in the manuscript, the nucleation and growth kinetics are coupled: the binding energy governs the potential for the nucleation of oxide embryo, while the fast diffusion along planar defects guarantee the subsequent oxide growth. Therefore, the steps near the TB or SF can facilitate successive oxide nucleation and growth (via the TB/SF nearby), while those away from the planar defect play negligible roles in oxidation, as demonstrated in Fig. 1.

3. The binding energy calculations were performed for interstitial O atoms on the free surface with different planar defects. However, the reaction between oxygen and metal atoms mainly occur at the metal/oxide interface during the oxide growth stage. I am wondering whether the trend of site-dependent binding energies for the metal/oxide system is similar to that for the Ag crystal only (free surface).

Response: We thank the reviewer for these insightful comments and agree with the reviewer that the reaction between oxygen and metal atoms mainly occur at the oxide-metal interface. In our experiment, an interface was formed after the nucleation of oxide at the free surface, which readily established an oxide-metal interface system. With the presence of TB or SF in metal, conspicuous inward oxidation along these planar defects was almost always observed (see Fig. 1, Fig. 3 and Supplementary Fig. 1-5). In contrast, extremely slow oxidation was observed after homogeneous oxide nucleation over the entire surface (see Supplementary Fig. 14). These different oxidation behaviours with and without planar defects after the formation of oxide-metal interface is schematically shown in Fig. R7. The rapid advancement of the oxide-metal interface in the presence of TB/SF gives solid evidence of site-dependent oxidation for the oxide-metal system. Moreover, the atomic model in Fig. 4 demonstrate an invariably lower CN at the TB tip intersecting with the oxide-metal interface, further demonstrating a general trend of defect-controlled binding energy.

Furthermore, despite several iterations, we have not been able to obtain satisfactory convergence on the *ab-initio* solutions of binding energy at an oxide-metal interface. Nevertheless, it is important to recognize that the *ab-initio* calculations presented in this work have required substantial computational resources, and that introducing a pre-existing oxide layer only added to the complexity of the simulations, without altering the main conclusions based on CN.

Fig. R7. Oxidation behaviours of metals with and without planar defects after the formation of oxide-metal interface (delineated with the red lines). The length of the arrows schematically demonstrates the different migration rate of the oxide-metal interfaces.

4. *Parallel TBs usually lead to a zig-zag surface structure in nanowires. The TB-surface junctions may have “convex” and “concave” configurations. The DFT calculation presented in this study mainly address the convex configuration. I am curious whether the concave junction has the similar binding energy as the convex one. If not, it may have different oxidation kinetics.*

Response: We thank the reviewer for this observation. Although our DFT simulations mainly focus on the TB-facilitated oxidation at the convex surface, similar oxidation dynamics (including both the preferential nucleation and the layer-by-layer growth) at the concave TB-surface junctions was also observed in our experiments, as shown in Supplementary Fig. 2. The physical origin of the preferential oxidation at the TB-surface junction, regardless of the convex or concave configuration, relates to the higher reactivity associated with lower coordination number (CN) of the surface atom at TB/SF junction. For our atomistic models of nanotwins in Fig. 2d (formerly Fig. 2e), CN is the smallest on atoms at the convex tip (CN=8) and largest at the concave V-shaped edge (CN=10), while CN=9 on those of the smooth (111) facet. In the concave configuration, however, it is worth mentioning that the V-edge at the TB-surface junction acts as the pinning site for diffusing Cu adatoms and promotes the dynamic formation of a Cu atom row (termed as W-chain) with lower CN and high chemical reactivity (Krajčí *et al.*, *J. Chem. Phys.* **145**, 084703 (2016)). Therefore, while the CN principle is consistent with the results of our theoretical calculations, certain local differences may exist experimentally at the concave and convex configurations due to dynamic surface atom reconstruction, which is beyond the scope of our current study.

5. *The mean Ag-O binding energies in Fig. 2 were calculated at different surface sites of nanotwinned Ag with varied TB spacing. How many surface site data were averaged for each TB spacing case? Besides, some atomic steps may exist on the nanotwinned crystal surface, which could also depend on the TB spacing as reported in Ref. 12. How would the pre-existed atomic steps affect the calculations?*

Response: The different configurations used for determining the mean O binding energies with varied TB spacings in Fig. 2d are all provided in Supplementary Figure 8. The mean values have been determined from the average of the two highest energies calculated for each model. Furthermore, in Fig. 2d, we have deliberately simulated only atomically smooth surfaces in order to decouple the intrinsic TB spacing and surface steps effects. The binding energy of surface steps is addressed separately in Fig. 2c. We find it relatively close to that at a convex TB tip. Therefore, we can only assume that adding surface steps to the twinned surfaces would increase the predicted mean O binding energies of planar faults.

Response to Reviewer #3

In the study by Zhu et. al., authors investigate the mechanisms and kinetics of selective defect driven oxidation of nanocrystalline metals using in-situ TEM characterization and DFT computations. The results of the study are novel and of general interest to the materials science community. The conclusions are based on strong, technically sound evidence. I recommend this manuscript for publication if authors can answer the following questions:

Response: We sincerely thank the reviewer for recognizing the novelty and generality of our work.

1) What is the source of the large error-bars in Fig. 2e, especially, in the case of SGL and certain spacings of TBS? While the mean binding energy is higher for TBS, and further increases with increasing spacing, does this trend still hold for the range of binding energies computed for each case, given the large error bars in some cases?

Response: The mean values have been determined from the average of the two highest energies calculated for each model. However, the error bars were erroneously calculated from the standard deviation of those two values, whereas using the half range reduced the error noticeably. We have updated the error bars in Fig. 2d in the revised manuscript (shown below as Fig. R8).

Furthermore, we have found that one of our DFT calculations for TBS = 7.2 Å did not properly converge to the global energy minimum and stopped on a local minimum, leading to a lower binding energy for one site. We have repeated this calculation by using a larger energy cutoff of 500 eV, which yielded better energy convergence and increased the binding for this site. Therefore, we have corrected this energy in Supplementary Figure 8 (shown below as Fig. R9). The associated mean energy and error bar at TBS = 7.2 Å now follows the trend as a function of TB spacing. The revised Fig. 2d (Fig. R8 below) shows that the trend holds well and the error is smaller.

Fig. R8. Mean E_b at different surface sites of SGL, TB and SF in Ag nanocrystals.

Fig. R9. Site-specific O binding energies calculated by ab initio simulation of different twin boundary configurations with spacings of (a) two (111) atomic layers (i.e., extrinsic stacking fault configuration), (b) three (111) atomic layers, (c) four (111) atomic layers, (d) six (111) atomic layers and (e) twelve (111) atomic layers.

2) While the authors have sufficiently discussed the mechanisms and kinetics of defect assisted nucleation and growth, can they comment on the thermodynamics of nucleation? Especially how does binding energy affects critical radius of nuclei that can spontaneously grow?

Response: We thank the reviewer for this constructive question, which inspired us to gain a more insightful understanding on the site-selective oxidation. From the thermodynamic perspective, the formation of oxide on the Ag surface is governed by the change of free energy, which can be derived as follow:

$$\Delta G = \Sigma A_s \gamma_s + \Sigma A_i \gamma_i + \Delta G(\text{Ag}_2\text{O}) - \Sigma A_{\text{Ag}} \gamma_{\text{Ag}}$$

where γ_s , γ_i and γ_{Ag} represent the surface energies of the oxide, oxide-metal interface and the Ag matrix; A_s , A_i and A_{Ag} denote the corresponding surface/interface area; ΔG (Ag_2O) is the volume energy of Ag_2O . It is widely acknowledged that the chemisorbed layer on the fresh metallic surface reduces the corresponding surface energy (Zhou and Yang *Surf. Sci.* **559**, 100–110 (2004)), resulting in a decreased value of $\Sigma A_{Ag}\gamma_{Ag}$. Therefore, the adsorption of oxygen atoms to the surface in fact impairs the intrinsic dynamics for further oxidation. However, the higher oxygen binding energies at the TB/SF (especially under low oxygen pressure) results in a site-selective oxygen adsorption, where localized oxide embryo and oxide-metal interface forms, in stark contrast to the large area interface over the entire surface, see Fig. R10 below). The energy rise associated with the oxide surface energy and the oxide-metal interface energy can be significantly reduced, leading to a much lower ΔG . Therefore, our qualitative thermodynamic analysis indicates that further oxidation process at room temperature should be essentially accelerated at the TB/SF compared with that at the flat surface without planar defects, which is consistent with our experimental observations (Fig. 1 and 3 and Supplementary Fig. 14).

We have added the above thermodynamic analysis of the nucleation and growth of the oxide in Discussion of the revised manuscript and Fig. R10 in the revised SI as Supplementary Figure 15, following the reviewer’s suggestion.

Fig. R10. Oxidation behaviours of metals with and without planar defects after the formation of oxide-metal interfaces (delineated with the red lines).

3) What is the physical reason behind higher binding energy of interstitial sites closer to twin/surface junctions and kink steps/sites near SFs or SFs/surface junction than sites/kink steps away from SFs and twins? Is it related to CN? However, I think authors state that “that the

coordinate number (CN) of atoms at the TB-surface junction (CN=8) is larger than that at the edge of a (111) kink step (CN=7)."

Response: To clarify, we are concluding that the lowest CN is a good indicator only when comparing binding sites of the same surface and configuration, but this is not the case when comparing sites across different configurations. Evidence for this can be found when comparing the CNs between the twinned tip (Fig. 2b) and the kink step (Fig. 2c), as noted by this reviewer, as well as the CN of single-crystal and stacking-faulted configurations in Fig. 2e (now revised Fig. 2d). In addition to CN, it is possible to hypothesize that other physical effects such as TB-induced surface strains (see for example, Deng and Sansoz, *Appl. Phys. Lett.* **95**, 091914 (2009)) could play a role on the surface binding energy. We have not quantified those strains in the present work.

4) Why does binding energy decrease with increasing twin density?

Response: We could attribute this decreased binding energy to the local change in the bonding environment of the oxidation site at the surface. Specifically, we have found that the oxidation sites with neighboring atoms of lower CN tend to possess higher binding energy. Atoms at the convex tip and on the neighboring (111) surfaces have a CN of 8 and 9, respectively, while those at the concave V-edge TB site have a larger CN of 10. Increasing the twin density decreases the distance between the V-edge site and the tip site. When the distance between two types of sites are small (*i.e.*, low TB spacing), the increased average CN among the neighboring atoms at the tip will reduce the binding energy. Such effects on the average CN and bonding environment become substantial only when the V-edge site is close enough to the tip, which explains the negligible change of E_b from TBS = 28.8 Å to 14.4 Å followed by considerable drop with decreasing TBS thereafter. We have added this discussion to the revised main text.

5) Authors state that "In contrast to nanotwinned Ag, however, we found that homogeneous oxidation initiates from multiple sites simultaneously in defect-free Ag single crystals (Supplementary Fig. 13); such process is mainly dominated by surface diffusion rather than defect-assisted diffusion, leading to significantly slower kinetics and reduced inward growth." If the oxidation in pure Ag is dominated by surface diffusion rather than defect-assisted diffusion, shouldn't oxidation kinetics be faster in pure case, since surface has the highest diffusivity than most defects? Authors should be careful while making such statements.

Response: We thank the reviewer for pointing out this mistake. The growth of oxide mainly involves the lattice diffusion of oxygen in the oxide, rather than surface diffusion. We have therefore changed the statement of "such process is mainly dominated by surface diffusion" into "such process is mainly dominated by conventional bulk diffusion" according to the reviewer's kind remark.

REVIEWERS' COMMENTS

Reviewer #1 (Remarks to the Author):

The authors have carefully addressed my concerns and comments. I would recommend for acceptance.

Reviewer #2 (Remarks to the Author):

My previous inquires and comments have been addressed satisfactorily in the revised manuscript. I will be happy to recommend the publication of this paper in Nature Communications.

Reviewer #3 (Remarks to the Author):

The authors have sufficiently addressed in the updated version all previous concerns and comments raised. I can recommend the paper for publication.